# Synthesis and Properties of New 3-Heterylamino-Substituted 9-Nitrobenzanthrone Derivatives

**DOI:** 10.3390/molecules28135171

**Published:** 2023-07-02

**Authors:** Armands Maļeckis, Marija Cvetinska, Aleksandrs Puckins, Sergejs Osipovs, Jelizaveta Sirokova, Sergey Belyakov, Elena Kirilova

**Affiliations:** 1Institute of Technology of Organic Chemistry, Faculty of Materials Science and Applied Chemistry, Riga Technical University, P. Valdena Str. 3, LV-1048 Riga, Latvia; 2Department of Applied Chemistry, Institute of Life Sciences and Technology, Daugavpils University, LV-5401 Daugavpils, Latvia; 3Latvian Institute of Organic Synthesis, Aizkraukles Str. 21, LV-1006 Riga, Latvia; serg@osi.lv

**Keywords:** benzanthrone, heterocycle, substituted amines, nitro derivative, fluorescence, solvatochromism, crystal structure

## Abstract

In the present study, new fluorophores based on disubstituted benzanthrone derivatives were designed starting from 9-nitro-3-bromobenzanthrone with nucleophilic substitution of the bromine atom with some secondary cyclic amines. It has been found that this reaction is positively affected by the presence of a nitro group in comparison with 3-bromobenzanthrone. The new compounds exhibit intense absorption and pronounced luminescent properties in various organic solvents. In this regard, their photophysical properties were evaluated with an experimental study of the solvatochromic behavior of the obtained compounds in various solvents. It has recently been found that the addition of an electron-withdrawing nitro group to the benzanthrone core increases its first- and second-order hyperpolarizability. Such dyes can be used in the fabrication of optical limiter devices. Therefore, the developed fluorescent molecules have a potential prospect for extensive application in optoelectronics.

## 1. Introduction

Fluorescence is known as a phenomenon where a substance absorbs light and promptly emits it at a longer wavelength—the process taking place within nanoseconds. The emitted light has a lower energy compared to the absorbed light, which results in a spectral shift towards longer wavelengths [1]. There are several characteristics that make an organic compound a good fluorophore. Such a compound should have a high absorption coefficient—the more efficiently the molecule can be excited, the brighter the emitted fluorescence [2]. A Stokes shift, the difference between the excitation and emission wavelengths, should be large enough to minimize reabsorption and maximize sensitivity [3]. An efficient fluorophore should also have a high quantum yield, indicating that most of the absorbed energy is converted into emitted light rather than the absorbed energy being transformed into other competing non-radiative processes and should be photostable so that it will not degrade or lose its fluorescent properties over time or with repeated exposure to light [4,5].

In the anthrone family of fluorescent dyes, a four-cyclic condensed aromatic ketone, benzanthrone, has been confirmed to display the excellent above-mentioned characteristics, which prompts the synthesis of new derivatives and their study [6,7]. Previously, benzanthrone compounds have found utilization as fluorescent bioimaging probes, which can aid in the visualization of parasitic trematodes and nematodes [8,9,10], as well as in the identification of callus embryos of different plant species through the use of confocal laser scanning microscopy imaging [11]. Moreover, these substances can be selectively deployed to identify amyloid fibrils [12,13]. Benzanthrone derivatives also have the potential to be utilized in liquid crystal displays [14,15], polymeric materials [16,17] and as probes to determine the pH levels and presence of cations in solutions [18,19,20]. These multiple benefits of these compounds have shown them to be a potential tool for various industries and research fields.

Previous studies have shown a significant effect of the substituent in the third position of the aromatic system on the photophysical properties of benzanthrone dyes [21,22,23]. This is explained by the ability of benzanthrone compounds to form a state based on the intramolecular charge transfer (ICT) between the electron-donating substituent and the electron-withdrawing carbonyl moiety, which leads to a significant charge redistribution upon excitation and, as a consequence, to pronounced fluorosolvatochromism.

The influence of such electron-donating groups as amino, amidino, alkoxy and other groups is described in the literature, showing the highest ICT character of the first transition for derivatives with the strongest donor group [22].

Over the last few decades, 3-aminobenzanthrone derivatives (including alkylamines and imines [6,15,24], amides [25,26,27], amidines [28] and aminophosphonates [29]) have become the subject of significant attention. Among these many derivatives, substituted 3-piperazinyl derivatives of benzanthrone [30,31], as well as 3-(4-(diphenylamino)phenyl)benzanthrone and perylenediimide–benzanthrone dyads, were recently found to exhibit nonlinear optical (NLO) properties [32,33], which propels their design and use not only for imaging and sensing but also for applications in laser technology and optical communications.

In the latest research, it has been found that the introduction of a nitro group at position 9 of the benzanthrone core increases the efficiency of the charge transfer, which results in a stronger NLO response [33]. Thus, considering all of the information mentioned above, we have chosen to share our knowledge on previously unreported nitrated benzanthrone derivatives. In this paper, we detail the synthesis of these newly obtained compounds and provide a comprehensive comparison of their photophysical properties.

## 2. Results and Discussion

### 2.1. Synthesis

The first representative of benzanthrone derivatives with a nitro group and a substituted amino group (morpholine residue) was previously synthesized and showed interesting optical properties [34]. Therefore, we continued the study of such substances using a similar synthesis technique to obtain them. The target compounds were synthesized with the reaction of the nucleophilic aromatic substitution of aryl bromide with an addition–elimination mechanism in the previously obtained 3–bromo-9-nitrobenzanthrone (**1**) by heating with an appropriate heterocyclic secondary amine in 1-methyl-2-pyrrolidone as a solvent (see Figure 1). The starting nitro derivative **1** was obtained with nitration of 3-bromobenzanthrone according to the procedure described in [35].

In contrast to the analogous nucleophilic substitution reaction in 3-bromobenzanthrone, the bromine atom in nitro derivative **1** is replaced faster and at a lower temperature. This is obviously due to the strong electron-withdrawing effect of the nitro group, as a result of which the electron density at the third carbon atom is significantly reduced and the attack of the nucleophile is facilitated. Therefore, the yields of target products **2**–**5** are higher than those of similar derivatives that do not contain a nitro group. The obtained substances are deeply colored dark red solids and have an intense luminescence in solutions.

### 2.2. NMR Spectra Analysis

Structures of the obtained compounds were confirmed with ^1^H and APT NMR spectra (see Appendix A); chemical shifts for tertiary carbon atoms and attached hydrogens were assigned on the basis of COSY and HSQC spectra (see Appendix A, Table 1). 

It was of interest to compare the chemical shifts of protons and carbon atoms in monosubstituted amino derivatives and nitrated amines in order to evaluate the effect of the nitro group on the NMR spectra of the substances under study.

For comparison, a pair of derivatives with a piperidine residue containing a nitro group (**5**) and a previously synthesized non-nitrated derivative (**6**) were analyzed. In the example of benzanthrone derivative **5**, as indicated by ^1^H and COSY NMR spectra, signals of H-C(1) and H-C(2) appear as doublets at 8.27 and 7.12 ppm, respectively, with a coupling constant of 8 Hz. H-C(4) (doublet at 8.48 ppm) and H-C(6) (doublet at 8.70 ppm) are not chemically equal, and while the signal of H-C(5) in the ^1^H NMR spectrum shows up as an apparent triplet, it is in fact a masked doublet of doublets with similar splitting; COSY NMR spectra confirms coupling of both H-C(4) and H-C(6) with H-C(5). Deshielded H-C(8), situated next to both the carbonyl group and nitro group, appears as a doublet downfield at 9.13 ppm. It is observed that there is coupling of H-C(8) with H-C(10), which is attested with the COSY NMR spectrum and equal *J* values of 2.5 Hz. H-C(10) is also split by neighboring H-C(11) (doublet at 8.20 ppm), which makes H-C(10) appear as a doublet of doublets. The same patterns are applicable to the rest of the synthesized derivatives **2**–**4**. 

Multiplicity and chemical shifts of the obtained compound **5** can be contrasted with previously reported non-nitrated compound **6**. It is noteworthy to mention that, as validated with HMBC NMR spectra, while H-C(6) is positioned downfield relative to H-C(4) for compound **5**, the opposite is true for non-nitrated compound **6**. Moreover, besides an additional signal of H-C(9), there is a change in relative position and multiplicity for H-C(10), both of which are masked doublets of doublets of doublets that appear as triplets in the ^1^H NMR spectrum. Obtained results fully correlate with previous NMR studies of benzanthrone derivatives [36,37,38].

### 2.3. X-ray Crystallographic Study

Synthesized compound **3** crystallizes from dichloromethane in the form of dark red crystals, the structure of which was determined in this work with an X-ray diffraction analysis of single crystals (see Figure 1 and Appendix A). 

Obtained compound **5** crystallizes from toluene in the form of red luminescent crystals, the structure of which was determined with an X-ray diffraction analysis of single crystals in this research (see Appendix A). 

A characteristic feature of the crystal structure of **5** is the fact that there are two independent molecules (A and B) in the asymmetric unit (see Figure 2). These molecules are slightly distinguished by the conformation (the rotation of the piperidine cycles relative to the benzanthrone systems). The torsion angles of C2–C3–N18–C19 are equal to −16.0(1) and −20.3(1)° for molecules A and B, respectively. In both molecules, the piperidine cycles are a chair conformation. The nitro groups of these molecules lie almost in the planes of the benzanthrone systems. 

Molecules are connected to each other through weak intermolecular bifurcated hydrogen bonds of the CH···O type. Oxygen atom O24(A) forms C1(B)–H1(B)···O24(A) and C11(B)–H11(B)···O24(A) hydrogen bonds with lengths of 3.293(2) and 3.314(2) Å (H1(B)···O24(A) = 2.38 Å, C1(B)–H1(B)···O24(A) = 160°; H11(B)···O24(A) = 2.41 Å, C11(B)–H11(B)···O24(A) = 159°). In turn, O24(B) forms hydrogen bonds C1(A)–H1(A)···O24(B) (with parameters C1(A)···O24(B) = 3.368(2) Å, H1(A)···O24(B) = 2.47 Å, C1(A)–H1(A)···O24(B) = 157°) and C11(A)–H11(A)···O24(B) (C11(A)···O24(B) = 3.242(2) Å, H11(A)···O24(B) = 2.32 Å, C11(A)–H11(A)···O24(B) = 164°). By means of these hydrogen bonds, molecular chains (bands) are formed in the crystal structure, approximately parallel to the crystallographic plane ($\bar{2}$ 2 1). 

In the crystal structure, π–π stacking interactions between benzanthrone systems are observed. Due to these interactions, the molecular stacks are formed in the crystal lattice. The rows of these stacks are arranged parallel to the crystallographic direction [1 $\bar{1}$ 0]. Each stack contains both molecules A and molecules B. Figure 3 shows such a stack. The shortest intermolecular atom–atomic contacts in the stacks are C9(A)···C8(B) (3.361(2) Å) and C4(A)···C10(A) (3.431(2) Å). 

### 2.4. Spectroscopic Properties

The synthesized dyes exhibit pronounced luminescent properties in various organic solvents. In this regard, the photophysical properties of the obtained derivative were evaluated, recording the absorption and emission spectra in the seven organic solvents with a wide range of polarities (see Appendix A).

The obtained spectral data are summarized in Table 2, Table 3 and Table 4 in comparison with the characteristics of the previously studied unnitrated 3-piperidinobenzanthrone (**6**) and 3-pyrrolidinobenzanthrone (**7**) (see Figure 2). The photophysical properties of morpholine derivatives have been analyzed in a recent study [34].

Amines **4** and **5** absorb light at 450–480 nm, while derivative **3** with a pyrrolidine fragment has a longer wavelength absorption band at 525–560 nm and also exhibits a larger bathochromic shift from nonpolar benzene to polar DMSO (33 nm) than derivatives **4** and **5** (18–19 nm). The pyrrolidine derivative **7** obtained earlier [39] exhibits absorption in the longest wavelength range compared to all other studied compounds, both monosubstituted and disubstituted. It is known that the main process that determines the photophysical properties in substituted amino derivatives of benzanthrone is the transfer of electron density from the amino nitrogen to the benzanthrone ring, the degree of which may also depend on other substituents [21,22,23]. 

Obviously, in the case of compound **7**, there is a stronger interaction between the donor and acceptor groups, which leads to a lower electronic transition energy and an increase in the charge transfer upon absorption of a light quantum. The addition of an electronegative nitro group to the molecule of compound **3** leads to competition between this substituent and the carbonyl group of the molecule and, consequently, to a new electron density distribution in the ground state. The hypsochromic shift of the absorption band in derivative **3** and the low sensitivity of the absorption maxima to the polarity of the solvent indicate a decrease in the ICT character of the electronic transition.

Compared to unnitrated amines, compounds **3**–**5** have more intense absorption. For the synthesized compounds, the bathochromic shift of luminescence maxima (from benzene to DMSO) is comparable to the bathochromic shift of unnitrated derivatives and is 60–85 nm.

Nitrated compounds **3** and **5** show a bathochromic shift of absorption maxima by 15–35 nm compared to non-nitrated derivatives **7** and **6**, but their emission maxima demonstrate a hypsochromic shift by 3–14 nm relative to the luminescence spectra of monosubstituted derivatives. As a result, the Stokes shifts of the spectra of nitro derivatives become smaller than those of non-nitrated derivatives (see Table 4).

## 3. Materials and Methods

### 3.1. Materials and Basic Measurements

All reagents were of an analytical grade (Aldrich Chemical Company, Munich, Germany) and were used as received. The progress of the chemical reactions and the purity of products were monitored with TLC on silica gel plates (Fluka F60254, 20*10, 0.2 mm, ready-to-use), using C_6_H_6_-CH_3_CN (3:1) as an eluent, and visualization under UV light. Column chromatography on silica gel was carried out on Merck Kieselgel (230–240 mesh) with dichloromethane as an eluent. Melting points were determined on an MP70 Melting Point System apparatus and were not corrected.

^1^H-, COSY-, APT-, HMBC- and HSQC-NMR spectra were recorded on a Bruker Avance 500 MHz (Bruker Corporation, Billerica, MA, USA) in CDCl_3_ at an ambient temperature, using solvent peaks as the internal reference. Chemical shift (δ) values are reported in ppm. High-resolution accurate mass measurements were performed employing Orbitrap Exploris 120 (Thermo Fisher Scientific, 168 Third Avenue, Waltham, MA, USA) operating at the Full Scan mode at a 120,000 resolution. The IR spectrum was recorded on a Thermo Scientific Nicolet iS50 Spectrometer (ATR accessory; no. of scans: 64; resolution: 4; data spacing: 0.482 cm^−1^).

### 3.2. Synthesis and Characterization

General Procedure for synthesis of derivatives **2**–**5**: 

In a 25 mL round bottom flask, a mixture of 0.3 g (0.8 mmol) of 3-bromo-9-nitrobenzanthrone, 0.5 g of the corresponding heterocycle and 5 mL of 1-methyl-2-pyrrolidone was heated at 90–100 °C for 2–3 h. After cooling, a mixture of 5 mL of ethanol and 10 mL of water was added; the precipitate was filtered off and dried. The resulting solid was dissolved in dichloromethane and purified with column chromatography on silica gel 40/100 as an eluent using toluene.

*3-Morpholino-9-nitro-7H-benzo[de]anthracen-7-one* (**2**), Obtained as a red compound in a 58% yield with an m.p. of 229–230 °C. R*_f_* = 0.63 (eluent C_6_H_6_-CH_3_CN, *v/v* 3:1). ^1^H NMR (500 MHz, CDCl_3_) δ 9.19 (d, *J* = 2.4 Hz, 1H, (8)), 8.75 (d, *J* = 7.3 Hz, 1H, (6)), 8.55 (d, *J* = 8.3 Hz, 1H, (4)), 8.39 (d, *J* = 6.8 Hz, 1H, (10)), 8.38 (d, *J* = 6.5 Hz, 1H, (1)), 8.30 (d, *J* = 8.9 Hz, 1H, (11)), 7.76 (t, *J* = 7.8 Hz, 1H, (5)), 7.21 (d, *J* = 8.1 Hz, 1H, (2)), 3.98 (t, *J* = 4.5 Hz, 4H, (2′, 6′)), 3.21 (t, *J* = 4.5 Hz, 4H, (3′, 5′)). ^13^C NMR (126 MHz, CDCl_3_) δ 182.17 (C=O), 154.28 (C), 146.61 (C), 141.30 (C), 131.65 (CH, (4)), 130.86 (CH, (6)), 130.46 (C), 129.76 (C), 128.57 (C), 127.88 (C), 127.46 (CH, (1)), 126.82 (CH, (10)), 126.29 (CH, (5), 123.94 (CH, (11)), 123.72 (CH, (8)), 119.77 (C), 115.10 (CH, (2)), 67.08 (CH_2_, (2′, 6′)), 53.79 (CH_2_, (3′, 5′)). FTIR (neat): 655, 649, 709, 744, 756, 768, 794, 804, 833, 872, 903, 925, 945, 954, 979, 1024, 1040, 1052, 1067, 1081, 1092, 1126, 1157, 1179, 1212, 1249, 1278, 1303, 1319, 1361, 1385, 1407, 1439, 1460, 1477, 1506, 1569, 1582, 1595, 1646, 2885, 2991, 3054. ESI-FTMS: calculated for [C_21_H_16_N_2_O_4_ + H^+^]: 361.1183, found: 361.1181.*9-Nitro-3-(pyrrolidin-1-yl)-7H-benzo[de]anthracen-7-one* (**3**), Obtained as a red compound in a 63% yield with an m.p. of 257–258 °C. R*_f_* = 0.73 (eluent C_6_H_6_-CH_3_CN, *v/v* 3:1). ^1^H NMR (500 MHz, CDCl_3_) δ 9.21 (d, *J* = 2.6 Hz, 1H, (8)), 8.78 (d, *J* = 7.3 Hz, 1H, (6)), 8.65 (d, *J* = 8.4 Hz, 1H, (4)), 8.33 (dd, *J* = 8.9, 2.6 Hz, 1H, (10)), 8.26 (d, *J* = 8.7 Hz, 1H, (1)), 8.20 (d, *J* = 8.9 Hz, 1H, (11)), 7.63 (dd, *J* = 7.9 Hz, 1H, (5)), 6.83 (d, *J* = 8.7 Hz, 1H, (2)), 3.76–3.70 (m, 4H, (2′, 5′)), 2.09–2.03 (m, 4H, (3′, 4′)). ^13^C NMR (126 MHz, CDCl_3_) δ 133.24 (CH, (4)), 130.99 (CH, (6)), 128.92 (CH, (1)), 126.42 (CH, (10)), 124.07 (CH, (8)), 123.72 (CH, (5)), 123.11 (CH, (11)), 108.86 (CH, (2)), 53.48 (CH_2_, (2′, 5′)), 26.10 (CH_2_, (3′, 4′)). FTIR (neat): 661, 675, 696, 743, 761, 768, 796, 818, 841, 861, 875, 890, 921, 966, 1007, 1039, 1072, 1100, 1113, 1146, 1169, 1213, 1242, 1266, 1290, 1312, 1344, 1405, 1444, 1497, 1524, 1556, 1567, 1582, 1603, 1644, 2847, 2959, 3083. ESI-FTMS: calculated for [C_21_H_16_N_2_O_3_ + H^+^]: 345.1234, found: 345.1232.*3-(4-Methylpiperazin-1-yl)-9-nitro-7H-benzo[de]anthracen-7-one* (**4**), Obtained as a red compound in a 59% yield with an m.p. of 271–273 °C. R*_f_* = 0.02 (eluent C_6_H_6_-CH_3_CN, *v/v* 3:1). ^1^H NMR (500 MHz, CDCl_3_) δ 9.14 (d, J = 2.5 Hz, 1H, (8)), 8.72 (dd, J = 7.3, 1.4 Hz, 1H, (6)), 8.51 (dd, J = 8.3, 1.4 Hz, 1H, (4)), 8.34 (dd, J = 8.9, 2.6 Hz, 1H, (10)), 8.31 (d, J = 8.2 Hz, 1H, (1)), 8.23 (d, J = 9.0 Hz, 1H, (11)), 7.73 (dd, J = 8.3, 7.3 Hz, 1H, (5)), 7.18 (d, J = 8.2 Hz, 1H, (2)), 3.24 (t, J = 4.8 Hz, 4H, CH_2_), 2.71 (brs, 4H, CH_2_), 2.40 (s, 3H, CH_3_). ^13^C NMR (126 MHz, CDCl_3_) δ 182.26 (C=O), 154.66, 146.53, 141.43, 131.88 (CH, (4)), 130.81 (CH, (6)), 130.39, 129.75, 128.52, 127.93, 127.56 (CH, (1)), 126.80 (CH, (10)), 126.10 (CH, (5)), 123.89 (CH, (11)), 123.77 (CH, (8)), 119.33, 115.10 (CH, (2)), 77.28, 77.03, 76.77, 55.26, 53.35, 46.19. FTIR (neat): 402, 479, 505, 596, 654, 699, 749, 777, 828, 889, 925, 956, 1010, 1073, 1141, 1169, 1242, 1287, 1328, 1372, 1453, 1503, 1575, 1651, 2692, 2786, 2833, 2939, 3090. ESI-FTMS: calculated for [C_22_H_19_N_3_O_3_ + H^+^]: 374.1499, found: 374.1485.*9-Nitro-3-(piperidin-1-yl)-7H-benzo[de]anthracen-7-one* (**5**), Obtained as a red compound in a 60% yield with an m.p. of 251–252 °C. R*_f_* = 0.92 (eluent C_6_H_6_-CH_3_CN, *v/v* 3:1). ^1^H NMR (500 MHz, CDCl_3_) δ 9.13 (d, *J* = 2.5 Hz, 1H, (8)), 8.70 (d, *J* = 7.3 Hz, 1H, (6)), 8.48 (d, *J* = 8.3 Hz, 1H, (4)), 8.31 (dd, *J* = 8.9, 2.6 Hz, 1H, (10)), 8.27 (d, *J* = 8.2 Hz, 1H, (1)), 8.20 (d, *J* = 8.9 Hz, 1H, (11)), 7.71 (dd, *J* = 7.8 Hz, 1H), 7.11 (d, *J* = 8.2 Hz, 1H), 3.16 (s, 4H), 1.84 (p, *J* = 5.6 Hz, 4H), 1.67 (p, *J* = 5.8 Hz, 2H). ^13^C NMR (126 MHz, CDCl_3_) δ 182.30 (C=O), 156.09 (C), 146.32 (C), 141.57 (C), 132.21 (CH, (4)), 130.74 (CH, (6)), 130.23 (C), 129.76 (C), 128.44 (C), 128.05 (C), 127.70 (CH, (1)), 126.68 (CH, (10)), 125.85 (CH, (5)), 123.76 (CH, (11)), 123.76 (CH, (8)), 118.54 (C), 114.82 (CH, (2)), 54.93 (CH_2_, (2′, 6′)), 26.31 (CH_2_, (3′, 5′)), 24.39 (CH_2_, (4′)). FTIR (neat): 406, 452, 478, 530, 586, 619, 666, 697, 743, 774, 824, 888, 921, 952, 989, 1025, 1068, 1128, 1169, 1229, 1273, 1329, 1376, 1440, 1505, 1570, 1645, 2662, 2702, 2738, 2818, 2852, 2919, 3066, 3109, 3997. ESI-FTMS: calculated for [C_22_H_18_N_2_O_3_ + H^+^]: 359.1390, found: 359.1381.*3-(Piperidin-1-yl)-7H-benzo[de]anthracen-7-one* (**6**), Obtained from 3-bromobenzanthrone at 120–130 °C for 6–7 h as an orange compound in a 48% yield with an m.p. of 165–166 °C. R*_f_* = 0.94 (eluent C_6_H_6_-CH_3_CN, *v/v* 3:1). ^1^H NMR (500 MHz, CDCl_3_) δ 8.67 (d, J = 7.3 Hz, 1H, (8)), 8.46 (d, J = 8.3 Hz, 1H, (4)), 8.38 (d, J = 7.9 Hz, 1H, (6)), 8.21 (d, J = 8.1 Hz, 1H, (1)), 8.12 (d, J = 8.2 Hz, 1H, (11)), 7.65 (dd, J = 7.8 Hz, 1H, (5)), 7.58 (dd, J = 7.6 Hz, 1H, (10)), 7.38 (dd, J = 7.5 Hz, 1H, (9)), 7.06 (d, J = 8.0 Hz, 1H, (2)), 3.05 (brs, 4H, (2′, 6′)), 1.79 (p, J = 5.6 Hz, 4H, (3′, 5′)), 1.61 (brs, 2H, (4′)). ^13^C NMR (126 MHz, CDCl_3_) δ 184.15 (C=O), 153.93 (C), 136.60 (C), 133.16 (CH, (10)), 131.26 (CH, (4)), 130.24 (C), 129.76 (CH, (6)), 129.24 (C), 129.00 (C), 128.35 (C), 127.97 (CH, (8)), 127.16 (CH, (9)), 125.49 (CH, (5)), 125.14 (CH, (1)), 122.53 (CH, (11)), 120.97 (C), 114.88 (CH, (2)), 55.01 (CH_2_, (2′, 6′)), 26.46 (CH_2_, (3′, 5′)), 24.48 (CH_2_, (4′)). FTIR (neat): 410, 450, 473, 507, 581, 625, 653, 703, 772, 842, 878, 939, 961, 1027, 1060, 1101, 1168, 1206, 1277, 1375, 1463, 1511, 1573, 1643, 2668, 2704, 2737, 2808, 2847, 2930, 3064. ESI-FTMS: calculated for [C_22_H_19_NO + H^+^]: 314.1539, found: 314.1530.

### 3.3. Spectroscopic Measurements

The spectral properties of the investigated compound were measured in benzene (C_6_H_6_), chloroform (CHCl_3_), ethyl acetate (EtOAc), acetone, ethanol (EtOH), dimethyl sulfoxide (DMSO) and dimethylformamide (DMF) with concentrations of 10^−5^ M at an ambient temperature in 10 mm quartz cuvettes. All solvents were of a p.a. or analytical grade. The absorption spectra were obtained using the UV-visible spectrophotometer SPECORD^®^ 80 (Analytik Jena AG, Jena, Germany). The fluorescence emission spectra were recorded on a FLSP920 (Edinburgh Instruments Ltd., Edinburgh, UK) spectrofluorometer in the visible range 500–800 nm. 

### 3.4. Single Crystal X-ray Diffraction Analysis

Single crystals of C_21_H_16_N_2_O_3_ (**3**) were investigated on a Rigaku, XtaLAB Synergy, Dualflex, HyPix diffractometer. The crystal was kept at 140.0(1) K during data collection. Using Olex2 [40], the structure was solved with the ShelXT [41] structure solution program using Intrinsic Phasing and refined with the olex2.refine [42] refinement package using Levenberg–Marquardt minimization. Crystal data for **3** are as follows: orthorhombic, space group *Pbca* (no. 61), *a* = 14.4599(2) Å, *b* = 7.1736(2) Å, *c* = 29.6287(6) Å, *V* = 3073.4(1) Å^3^, *Z* = 8, *T* = 140.0(1) K, μ(Cu Kα) = 0.822 mm^−1^, *Dcalc* = 1.4884 g/cm^3^, 19,757 measured reflections (2Θ ≤ 160°) and 3342 unique (*R*_int_ = 0.0356, *R*_sigma_ = 0.0343) that were used in all calculations. The final *R*_1_ was 0.0430 (*I* > 2σ(*I*)) and *wR*_2_ was 0.1216 (all data). 

Diffraction data of compound **5** were collected at 150 K on a Rigaku, XtaLAB Synergy, Dualflex, HyPix diffractometer using CuKα radiation (λ = 1.54184 Å). The crystal structure was solved with direct methods [43] and refined using Gauss–Newton minimization with the help of a software package [42]. Crystal data for **5** are as follows: triclinic; a = 9.0593(1), b = 12.1264(2), c = 15.8953(2) Å, α = 87.867(1), β = 76.570(1), γ = 86.651(1)°; V = 1695.05(4) Å3, Z = 4, μ = 0.766 mm–1 and Dcalc = 1.404 g·cm–3; space group is P $\bar{1}$; R[F2 > 2σ(F2)] = 0.0417. For further details, see crystallographic data for **5** deposited at the Cambridge Crystallographic Data Centre as the Supplementary Publication Number CCDC 2,233,481. Copies of the data can be obtained, free of charge, on application to CCDC, 12 Union Road, Cambridge CB2 1EZ, UK.

## 4. Conclusions

In the present research, a synthetic method for preparing new disubstituted heterylaminobenzanthrones was implemented from 9-nitro-3-bromobenzanthrone. The synthesized derivatives were obtained with 59–63% yields as crystalline deeply colored substances with an intense luminescence in organic solvents.

The obtained compounds absorb at 450–560 nm with large extinction coefficients and emit at 570–660 nm. The results obtained indicate that emission of the aimed derivatives is sensitive to the solvent polarity showing positive fluorosolvatochromism.

Taking into account the fact that the addition of an electron-withdrawing nitro group to the benzanthrone molecule increases its first- and second-order hyperpolarizability, it can be assumed that the developed fluorescent compounds have a potential prospect for application in optoelectronics.

## Data Availability

Data are contained within this article or Appendix A.

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
