# Peer review of "Synthesis and Properties of New 3-Heterylamino-Substituted 9-Nitrobenzanthrone Derivatives"

_molecules, 2023, doi:10.3390/molecules28135171_

Round 1

Reviewer 1 Report

The manuscript entitled “Synthesis and properties of new nitroheterylaminoderivatives of benzanthrone” written by Maleckis and co-workers describes on synthesis of titled compounds and their physical properties. The reviewer cannot recommend this paper to be published in the present form. Please revise along points listed below.

1. Present structure of compounds 6 and 7.

2. Add numbering in Figure 1, compound 1. Paretheses are not required for number of compounds in this Figure.

3. Page 2 line 73; “nucleophilic substitution of bromine” should be revised as “nucleophilic aromatic substitution of aryl bromide by addition-elimination mechanism.”

4. Page 4; Figure 1;it is not compound 6 but compound 5.

5. Experimental: Add eluent to each Rf values.

6. Page5-6: Absorption of maxia ramda of compound 5 was longer than those of corresponding non-nitrated compound 6. This seems reasonable because conjugate system was expanded by addition of nitro group at aromatic group. But the maxia of the corresponding pyrrolidine compound 3 was went shorter than non-nitrated compound 7. How do the authors explain this? Please add some idea of the authors.

Author Response

Point 1: Present structure of compounds 6 and 7.
Response 1: As suggested by the reviewer, we prepared Scheme 2 with structure of compounds 6 and 7. Page 3.
Point 2: Add numbering in Figure 1, compound 1. Paretheses are not required for number of compounds in this Figure. Response 2: The numbering and other corrections were made to Scheme 1. Page 3. Point 3: Page 2 line 73; “nucleophilic substitution of bromine” should be revised as “nucleophilic aromatic substitution of aryl bromide by addition-elimination mechanism.” Response 3: Corrected. Lines 84-85.
Point 4: Page 4; Figure 1;it is not compound 6 but compound 5.
Response 4: We appreciate your thoroughness. Corrected.
Point 5: Experimental: Add eluent to each Rf values. Response 5: The eluent was added. Lines 250, 263,275, 287, 301.
Point 6: Page5-6: Absorption of maxia ramda of compound 5 was longer than those of corresponding non-nitrated compound 6. This seems reasonable because conjugate system was expanded by addition of nitro group at aromatic group. But the maxia of the corresponding pyrrolidine compound 3 was went shorter than non-nitrated compound 7. How do the authors explain this? Please add some idea of the authors
Response 6: Thanks to the reviewer for this suggestion. On the recommendation of the reviewer, we supplemented the discussion with additional information: “Pyrrolidine derivative 7 obtained earlier [39] exhibits absorption in the longest wavelength range compared to all other studied compounds, both monosubstituted and disubstituted. It is known that the main process that determines the photophysical properties in substituted amino derivatives of benzanthrone is the transfer of electron density from the amino nitrogen to the benzanthrone ring, the degree of which may also depend on other substituents [21-23].
Obviously, in the case of compound 7, there is a stronger interaction between the donor and acceptor groups, which leads to a lower electronic transition energy and an increase in charge transfer upon absorption of a light quantum. The addition of an electronegative nitro group to the molecule of compound 3 leads to competition between this substituent and the carbonyl group of the molecule
and, consequently, to a new electron density distribution in the ground state. The hypsochromic shift of the absorption band in derivative 3 and the low sensitivity of the absorption maxima to the polarity of the solvent indicate a decrease in the ICT character of the electronic transition.” Lines 193-206.
Thank you so much for taking the time to review our manuscript. We are grateful to the reviewers for the constructive comments

Reviewer 2 Report

Review Report

1.     A brief summary:

This manuscript provides a characteristic description of several 9-hetrerylamino-substituted 9-nitrobenzanthrone derivatives. The described compounds were synthesized in two steps by first nitration of the starting 3-bromobenzonthrone and subsequent modification of the obtained nitro derivative into a nucleophilic substitution reaction with some secondary cyclic amines. All compounds were obtained in moderate yields, their structures were confirmed by 1H-NMR, 13C-NMR and HRMS analyses, and characterized by IR spectroscopy. The photophysical properties of their solutions in various solvents were studied by spectrophotometry and fluorescent spectrometry. The crystallographic description of the structure was performed for one compound. The influence of the nitro group on the reactivity, the shift of the atomic signals in the NMR spectra, as well as the absorption and fluorescence spectra, is discussed.

2.     Questions and recommendations:

2.1  The name “nitroheterylaminoderivatives” is not quite correct in the title of the article. I think it's better to use the following options: “Heterylamino-substituted 9-Nitrobenzanthrone derivatives” or “3-Heterylamino-9-nitro benzanthrones”.

2.2  Please check the correctness of reference [28] in the “Synthesis” section, line 72. It should probably be reference [33].

2.3  In the "Introduction" section, justification should be added for the choice of secondary amines as substituents. What effect did you expect to achieve? Perhaps literature should be added that describes the luminescent properties of amino-substituted compounds.

2.4  Slightly less than 50% (19 out of 39) of the cited articles belong to the authors of this manuscript. Perhaps this is an excessive self-citation and more references to the work of other authors should be added or the introduction should indicate that a limited number of research groups are investigating the properties of benzanthrone derivatives.

2.5  The use of the term “nitroamines” (line 295 in “Conclusions”) is not correct, since the article is not about nitroamines, but about derivatives containing a nitro group in one position and a heteroamino group in another position of benzanthrone.

2.6  The article discusses the comparison of the obtained nitro derivatives with analogues without a nitro group. We are talking about connections numbered 6 and 7. Connection 6 is described, but connection 7 is not. Perhaps it is described in some already published article, then you need to provide this reference.

2.7  If the article discusses the comparison of the properties of the obtained compounds, then in tables 2-4 it is necessary to add data on compound 2.

2.8  It is better to add data on compound 7 to table 1, since data on it is available in other tables.

2.9  Why are the data in the tables not in the order of connections: 6,5,3,7,4? It would be much better to present the results in order. Also, the tables look very “simple”. Combined rows should be added to all tables with the name of the quantity, the values of which are given in the columns below. For example, like this:

It is also better to remove horizontal borders in tables.

2.10 In section 3 “Materials”, the description of instruments for studying the spectroscopic properties of substances should be given in one section. the last paragraph of paragraph 3.1 and the content of paragraph 3.3 duplicate each other.

3.     Typos:

3.1  In the title of Table 1, as well as in the titles of the columns, the compound numbers are indicated in italics. The same in lines 112, 113, 115.

3.2  In line 97, connection number 5 should be in bold type.

3.3  Line 193 misspelled the name of the chemical equipment manufacturer. Spelled “Fisher”, not “Fischer”.

3.4  The names of solvents in all tables should be centered, as well as other inscriptions.

Author Response

Point 2.1: The name “nitroheterylaminoderivatives” is not quite correct in the title of the article. I think it's better to use the following options: “Heterylamino-substituted 9-Nitrobenzanthrone derivatives” or “3-Heterylamino-9-nitro benzanthrones”.

Response 2.1: As suggested by the reviewer, we corrected the title from “Synthesis and properties of new   nitroheterylaminoderivatives of benzanthrone” to “Synthesis and properties of new  3-heterylamino-substituted 9-nitrobenzanthrone derivatives”. Lines 2-3

Point 2.2: Please check the correctness of reference [28] in the “Synthesis” section, line 72. It should probably be reference [33].

Response 2.2: We appreciate your thoroughness. Corrected to [34]. Line

Point 2.3: In the "Introduction" section, justification should be added for the choice of secondary amines as substituents. What effect did you expect to achieve? Perhaps literature should be added that describes the luminescent properties of amino-substituted compounds.

Response 2.3: Following the recommendation of the reviewer, we supplemented the introduction with additional literature data on amino-substituted compounds: “Previous studies have shown a significant effect of the substituent in the third position of the aromatic system on the photophysical properties of benzanthrone dyes [21-23]. This is explained by the ability of benzanthrone compounds to form a state based on intramolecular charge transfer (ICT) between the electron-donating substituent and the electron-withdrawing carbonyl moiety, which leads to a significant charge redistribution upon excitation and, as a consequence, to pronounced fluorosolvatochromism.

The influence of such electron-donor groups as amino, amidino, alkoxy and other groups is described in the literature, showing the highest ICT character of the first transition for derivatives with the strongest donor group [22].”

Point 2.4:  Slightly less than 50% (19 out of 39) of the cited articles belong to the authors of this manuscript. Perhaps this is an excessive self-citation and more references to the work of other authors should be added or the introduction should indicate that a limited number of research groups are investigating the properties of benzanthrone derivatives.

Response 2.4: References to the works of other authors were added to the manuscript.

Point 2.5: The use of the term “nitroamines” (line 295 in “Conclusions”) is not correct, since the article is not about nitroamines, but about derivatives containing a nitro group in one position and a heteroamino group in another position of benzanthrone.

Response 2.5: "Nitroamines" have been adjusted for "derivatives". Line

Point 2.6: The article discusses the comparison of the obtained nitro derivatives with analogues without a nitro group. We are talking about connections numbered 6 and 7. Connection 6 is described, but connection 7 is not. Perhaps it is described in some already published article, then you need to provide this reference.

Response 2.6: The reference was addedPyrrolidine derivative 7 obtained earlier [38]” Line

Point 2.7:  If the article discusses the comparison of the properties of the obtained compounds, then in tables 2-4 it is necessary to add data on compound 2.

Response 2.7: Compound 2 data were added to Tables 2-4.

Point 2.8:  It is better to add data on compound 7 to table 1, since data on it is available in other tables.

Response 2.8: NMR data for compound 7 were published previously.

Point 2.9: It would be much better to present the results in order. Also, the tables look very “simple”. Combined rows should be added to all tables with the name of the quantity, the values of which are given in the columns below. For example, like this: It is also better to remove horizontal borders in tables.

Response 2.9: The results were reordered. The horizontal borders were removed. Unfortunately, the example suggested by the reviewer is not visible.

Point 2.10:  In section 3 “Materials”, the description of instruments for studying the spectroscopic properties of substances should be given in one section. the last paragraph of paragraph 3.1 and the content of paragraph 3.3 duplicate each other.

Response 2.10: The last paragraph of paragraph 3.1 was deleted.

All typos corrected.

Thank you so much for taking the time to review our manuscript. We are grateful to the reviewers for the constructive comments.

Round 2

Reviewer 1 Report

The manuscript was satisfactory revised along the reviewer’s suggestions. The, the reviewer recommends this paper to be published in Molecules.

Author Response

Thank you so much for taking the time to review our manuscript.

Reviewer 2 Report

Thank you, all comments are taken into account. Additions according to Remark 2.9:

Values should be added to the headers of all data tables.

Author Response

Tables 1-4 were corrected.

Thank you so much for taking the time to review our manuscript.